# Learning to Plan and Generate Text with Citations

## Abstract

The increasing demand for the deployment of LLMs in information-seeking scenarios has spurred efforts in creating verifiable systems, which generate responses to queries along with supporting evidence. In this paper, we explore the *attribution* capabilities of plan-based models which have been recently shown to improve the faithfulness, grounding, and controllability of generated text. We conceptualize plans as a sequence of questions which serve as *blueprints* of the generated content and its organisation. We experiment with two models that utilize different variants of blueprints, an *abstractive* model where questions are generated from scratch, and an *extractive* model where the decoder is forced to copy questions from the input. Experiments on long-form question-answering show that output quality improves for blueprint models when these learn to generate responses with attribution. Moreover, the citations generated by blueprint models are more accurate compared to those obtained from LLM-based pipelines lacking a planning component.

## 1 Introduction

Large language models (LLMs) have demonstrated remarkable abilities to engage in creative conversations (Thoppilan et al., 2022; OpenAI, 2023), summarize information from contextual cues (Goyal et al., 2022; Zhang et al., 2023), and deliver exceptional zero-shot performance on a wide range of previously unseen predictive and generative tasks (Brown et al., 2020; Chowdhery et al., 2022). They are also becoming increasingly useful in information-seeking scenarios, ranging from answering simple questions (Roberts et al., 2020; Rae et al., 2021) to generating long-form responses to search-like queries (Nakano et al., 2021; Menick et al., 2022).

The increasing demand for the deployment of LLMs in information-seeking scenarios has further spurred efforts in creating verifiable systems, which generate responses to queries along with supporting evidence. The evidence can take the form of a URL pointing to a short segment of text which supports an answer (Bohnet et al., 2022), an attribution report with evidence snippets (Gao et al., 2022), quotes cited verbatim from pages retrieved from a search engine (Menick et al., 2022), and references to passages extracted while browsing (Nakano et al., 2021; Gao et al., 2023). In fact, this last type of evidence has been recently adopted in the form of in-line citations by commercial search engines such as BingChat[1] and perplexity.ai[2].

Regardless of how the evidence is presented, recent approaches tend to rely on a retrieval system (e.g., a commerical search engine) to obtain passages relevant to a query, while an LLM conditions on them to generate a response (Menick et al., 2022; Nakano et al., 2021; Bohnet et al., 2022). Other work generates an answer to the input query first and subsequently retrieves relevant evidence in a post-processing step (Bohnet et al., 2022). Alternatively, the retrieved evidence can be used to further revise the generated response rendering it more consistent with the evidence (Gao et al., 2022).

Despite recent efforts, it remains an open question how to best develop models with a built-in mechanism for attribution to external evidence. A related question is whether said mechanism contributes to generating more factually faithful output. Large-scale evaluation studies paint a worrying picture. Liu et al. (2023) find that long-form responses obtained from existing search engines frequently contain unsupported statements or inaccurate citations, while Bohnet et al. (2022)

---

[1] https://bing.com/new
[2] https://perplexity.ai

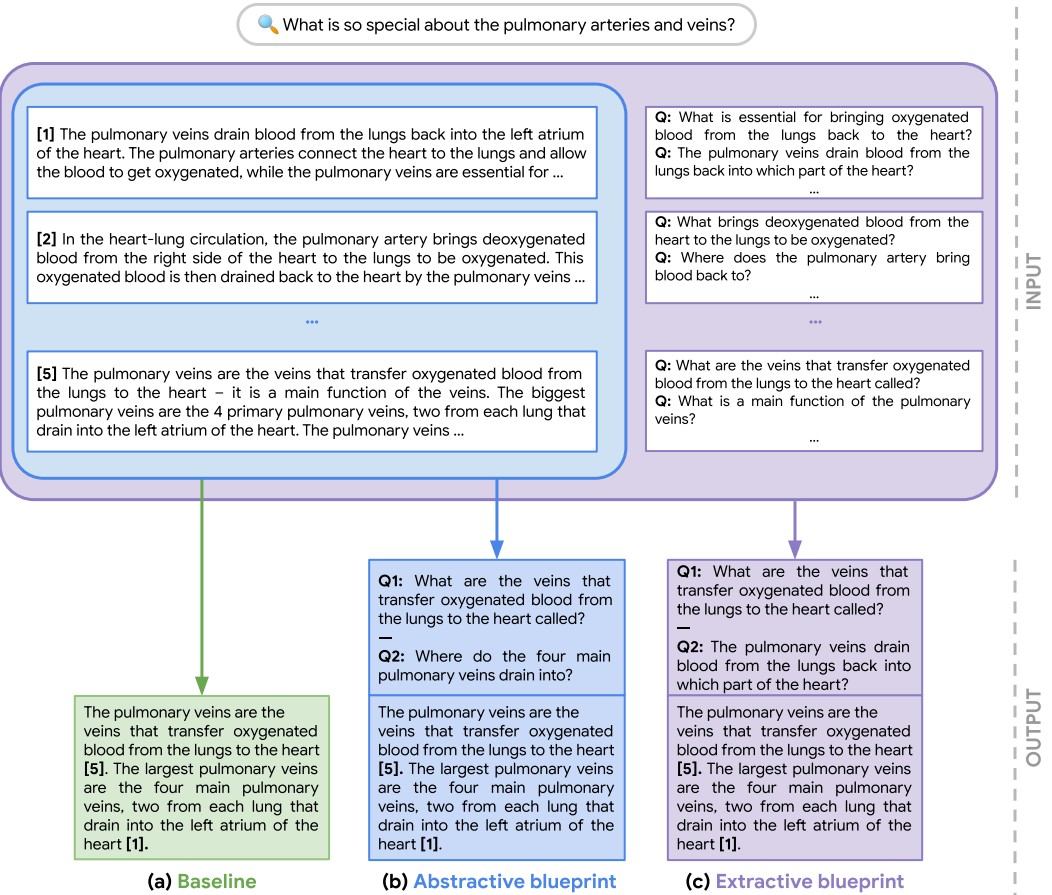

Figure 1: Example query (top), followed by most relevant (abridged) passages, and output summaries (bottom) with in-line citations. Summary (a) is the output of a vanilla sequence-to-sequence model trained to generate long answers with citations. Summaries (b) and (c) are the output of blueprint models with abstractive and extractive plans, respectively. Citations for blueprint Models can have different formats, e.g., include references to the question plan or be implicit (see Section 5.2).

show that model performance on attribution varies greatly (between 46% and 71%) across different architectures for the simpler question answering task.

In this paper, we focus on long-form question answering (Fan et al., 2019), which shares many commonalities with query-focused summarization (Vig et al., 2022; Xu & Lapata, 2022). Both tasks aim to generate summaries or long answers from a set of passages that answer a specific query. We simulate how a search engine might synthesize passages of high relevance to a user query by assuming access to a retriever, and some way of verifying the output (e.g., by citing sources). Under this setting, we propose several model variants which operate on retrieved passages and learn to generate summaries with attribution (see Figure 1 for an example). We compare and contrast two sequence-to-sequence model families: on the one hand models which only generate the response (Figure 1(a)) and on the other, models which first predict an intermediate plan-based representation. Following Narayan et al. (2023), we conceptualize text plans as a sequence of questions operating as *blueprints* for generation, determining what to say and in which order (see Figures 1(b) and (c)).

Blueprint models offer several advantages over blackbox generation: their predictions can be examined, and traced back to the blueprint, questions are intuitive and can be used to modulate the content and length of the output. Moreover, on account of being more explainable, blueprint models allow for different forms of attribution, e.g., questions can be verified via citations to passages, while summaries can be verified through citations to the blueprint, passages, or both (see Figure 3).

In this paper, we explore how blueprint models can be extended to perform attribution, and whether an explicit planning mechanism has any bearing on citation quality. Specifically, our work investigates three important questions: (1) Does attribution as such yield better quality summaries (e.g., more informative or faithful) compared to modes which do not attribute to their sources; (2) Does the attribution mechanism matter (e.g., do plan-based models yield more accurate citations); (3) Once acquired, is attribution a robust skill or sensitive to task and domain changes (e.g., different question styles and QA tasks). To answer the first two questions, we use the AQuAMuSe dataset (Kulkarni et al., 2020) to train three different Transformer (Vaswani et al., 2017) models: one that does not use planning, an abstractive blueprint model where questions are generated by the model from scratch (see Figure 1(a)), and an extractive blueprint model where the decoder is forced to copy questions from the input (see Figure 1(b)). For the third question, we perform (out-of-domain) experiments on datasets representing different information seeking tasks and user requirements (for a fair comparison, all methods have access to the *same* retrieved passages).

Our results demonstrate that the use of a blueprint plan consistently improves attribution quality. Furthermore, we see significant gains in summary quality when using an extractive blueprint model. We also observe a tradeoff between abstractive and extractive blueprints, where the former better grounds the output to the plan while the latter better grounds the plan to the input. Finally, in terms of attribution quality, our models are competitive with (and sometimes better than) pipelines that heavily rely on large language models.

## 2 RELATED WORK

Large Language Models (LLMs) have made significant advances in recent years in generating high quality natural language responses (Brown et al., 2020; Chowdhery et al., 2022). Despite impressive capabilities, they also demonstrate high levels of opacity: it is unclear where the information they generate comes from, which can undermine their trustworthiness (Cheng et al., 2022).

A variety of techniques aim to remedy this by building language models that provide references to support generated text. The majority of existing work has focused on models which *learn* to generate citations. For example, Nakano et al. (2021) and Menick et al. (2022) use reinforcement learning from human preferences to train language models to answer questions and provide supporting evidence (in the form of retrieved snippets). Bohnet et al. (2022) use LLMs to generate both an answer and a url to a web page from which a paragraph is subsequently selected as support for the answer. They also experiment with attribution as a post-processing step where no learning is involved and use retrieval to select sources supporting their model's output. Along similar lines, Gao et al. (2022) propose a two-stage approach where generated text is first post-edited to be made attributable (to web content) and then revised accordingly. In addition to modeling, efforts have been also directed to the creation of evaluation protocols and benchmarks for assessing whether a statement is supported by provided evidence (Rashkin et al., 2021; Bohnet et al., 2022; Liu et al., 2023).

In our work we explore the attribution capabilities of plan-based models which have been shown to be less prone to hallucinations and more controllable (Moryossef et al., 2019; Puduppully et al., 2019; Narayan et al., 2021; 2023; Huot et al., 2023). The approach described in Narayan et al. (2023) uses a *blueprint* text plan, formulated as a sequence of question-answer pairs, to serve as an intermediate representation for content selection and organization of the generated text. We argue that blueprints are ideally suited to attribution, as the questions provide a natural link between retrieved passages and their summaries. We propose automatic methods for annotating training data with blueprints and citations and fine-tune several model variants to generate attributed text.

## 3 PROBLEM FORMULATION

We follow a formulation of query-focused summarization common in the literature (Xu & Lapata 2020; Vig et al. 2022; *inter alia*). Let $q$ denote an information seeking request (e.g., "What is so special about the pulmonary arteries and veins?" in Figure 1) and $\mathcal{P} = \{p_1, \ldots, p_n\}$ be a set of passages most relevant to $q$ (see top left in Figure 1). We adopt a two-step approach where $k$ most relevant passages are first retrieved and possibly reranked based on the query alone, and then fed to a secondary model which synthesizes the passages into a final summary $\mathcal{S} = \{s_1, \ldots, s_m\}$ consisting of sentences $s_i$. An *attributed* summary further includes supporting evidence. For the

sake of simplicity, we assume this is expressed by embedded in-line citations (see bottom in Figure 1 left). Each $s_i$ has a set of citations $\mathcal{C}_i = \{c_{i,1}, \ldots, c_{i,k}\}$ where $c_{i,j}$ is a passage ID signifying that sentence $i$ is citing the said passage. For the summary (a) in Figure 1, $\mathcal{C}_1 = \{[1]\}$, and $\mathcal{C}_2 = \{[2]\}$.

In order to generate text with attribution, we must somehow annotate target summaries with citations. We obtain silver-standard citations automatically, by measuring the entailment between passage $p_j \in \mathcal{P}$ as the premise and sentence $s_i \in \mathcal{S}$ as the hypothesis. We annotate $S_i$ with $c_{i,j}$ for the passage $p_j$ with the highest entailment score (see Section 4 for details on the entailment classifier we used). We next discuss how we train summarization models with attribution and also introduce blueprint variants thereof.

## 3.1 SEQUENCE-TO-SEQUENCE MODEL

Datasets for query-focused summarization and long-form question answering such as AQuAMuSe (Kulkarni et al., 2020) or ASQA (Stelmakh et al., 2022) typically consist of queries $\mathcal{Q}$, passages $\mathcal{P}$, and target summaries $\mathcal{S}$. A vanilla sequence-to-sequence model, generates summary $S$ given passages $P$ and query $q$, hence modeling the conditional probability $p(\mathcal{S}|P, q)$. As already explained (§3), in the case of attributed generation, the model is trained on summaries enriched with citations assumed to be part of the summary's token vocabulary. In Figure 1, the query, passages, and summary (a) with in-line references to passages [1] and [5] would constitute a training instance for this model.

## 3.2 BLUEPRINT MODELS

Narayan et al. (2023) formalize three types of blueprint models, which are purportedly suited to different generation tasks (e.g., long vs short output/input). We work with their end-to-end setup as it is a straightforward extension of the standard approach described above. The model encodes query $q$ and passages $\mathcal{P}$, and then generates, $\mathcal{B}; \mathcal{S}$, the concatenation of blueprint $\mathcal{B}$ and output summary $\mathcal{S}$ in one decoding step. We define $\mathcal{B}$ as a set of questions $B = \{b_1, \ldots, b_k\}$ that are answered in $\mathcal{S}$ and act as a proxy for content selection and planning. We depart from Narayan et al. (2023) in defining blueprints as *a set of questions* rather than question-answer pairs. This definition is more flexible, as we do not require answers to be extracted from passages, and are not limited to a specific style of answers or questions such as those manifested in SQuAD (Rajpurkar et al., 2018). Questions can be more general while answers can be abstractive, and represent a variety of syntactic constructs beyond noun phrases (e.g., verbs, clauses, even sentences). We assume the model has access to training data consisting of queries, passages, blueprints, and summaries (See Figure 1). Blueprints are not generally available, we explain how we obtain these automatically in the next section and define two blueprint models: an abstractive model (Figure 1 (b)) and an extractive model (Figure 1 (c)); these differ on whether the question-based plan is generated abstractively or copied from the input.

**Abstractive Model**  Following Narayan et al. (2023), we augment the training data with blueprints by generating questions for each target summary. We adopt an overgenerate-and-rank strategy, generating multiple questions which we then filter based on their overlap with the summary.

A challenge with blueprint models is that the output may be answering a mix of general and specific questions. To avoid generating only one style of questions, we employ two datasets originally developed for question answering, namely SQuAD (Rajpurkar et al., 2018) and Natural Questions (Kwiatkowski et al., 2019b). QA datasets typically consist of question/paragraph/answer triples but can be repurposed for question generation; to produce general-purpose questions, we assume the paragraph is the input and the question is the output, while for more specific questions, we concatenate the paragraph with the sentence where the answer span is found and the model's aim is to generate a question for the specific answer. We fine-tune T5-3B (Raffel et al., 2020) to obtain these two flavors of question generation models (see Appendix A for training details).

For each of the question generation models, we sample (through beam search) 10 questions per sentence and another 10 for the summary so as to have a diverse question set. For each summary sentence, we then select the question with the highest lexical overlap from the set of questions specific to that sentence *and* the set of summary-level questions. Blueprint questions are sorted according to the order of appearance of their corresponding summary sentences. Following Narayan et al. (2021), we add a special character (– in Figure 1) in the blueprint to separate the questions corresponding to

each sentence. We provide an example of how questions are generated and subsequently filtered in Appendix C.

**Extractive Model** In the extractive model, the questions align more closely with the input passages, which is useful in the context of information-seeking tasks. Specifically, we explicitly verbalize the questions each passage might answer, i.e., $\mathcal{P}_{ext} = \{(p_1, \mathcal{Q}_1), \ldots, (p_n, \mathcal{Q}_n)\}$, where $p_i$ are the retrieved passages and $\mathcal{Q}_i$ the corresponding passage questions. Blueprint $\mathcal{B}$ then consists of a subset of questions whose answers are found in the input. The extractive model is functionally equivalent to the abstractive one, the only difference being that it encodes passages $\mathcal{P}_{ext}$ (instead of $\mathcal{P}$). The extractive model also outputs $\mathcal{B};\mathcal{S}$, however, the decoder learns to *copy* questions from the input in order to construct $\mathcal{B}$. The extractive model has two advantages. Firstly, the questions are simply copied from the input, which reduces the risk of hallucinations and irrelevant information. Secondly, attribution comes for free, as we know which passage gave rise to which questions (see Figure 1). An obvious drawback is computational efficiency, as questions must be generated for every input passage.

We create a large number of questions for each passage using the same generation models described above. We then greedily select 5 questions ($\mathcal{Q}_i$) for each $p_i$ such that they have highest lexical overlap with the passage and minimal overlap with each other. Subsequently, we construct the target extractive blueprints under the assumption that their questions ought to be aligned to input passages *and* the output summary. We first discard any passage questions that cannot be answered in the target summary using an answerability classifier, a T5 model trained on a repurposed version of SQuAD v2 (Rajpurkar et al., 2018) that predicts whether a given question can be answered by a given passage (see Appendix B for training details). The remaining questions are then compared against the target summary and for each sentence we select a question based on lexical overlap. As in the abstractive model, a special character is also added to tell which blueprint questions correspond to which sentences. We provide an example of extractive blueprint selection in Appendix C.

## 4 EXPERIMENTAL SETTING

**Datasets** The bulk of our experiments use AQuAMuSe (Kulkarni et al., 2020), a query-focused summarization dataset, created with the intent of simulating how a search engine might synthesize documents of high relevance to a user query. It consists of queries taken from the Natural Questions (Kwiatkowski et al., 2019a) dataset, passages from Common Crawl, and multi-sentence summaries from Wikipedia. We use the same splits released in Kulkarni et al. (2020) which contain 6,599, 714, and 849 examples for training, development, and testing, respectively. The average query/response length is 9.2/107.3 words, and the number of input passages is 10.

All our models operate on the same passages which are reranked using T5-R (Huebscher et al., 2022), a T5 11B-based encoder trained with a classification loss on the MS MARCO dataset (Nguyen et al., 2016). T5-R reorders passages $p \in \mathcal{P}$ based on their relevance with query $q$. Target summaries in AQuAMuSe (Kulkarni et al., 2020) were annotated with in-line citations (see summaries in Figure 1) using an entailment classifier (T5-11B; Raffel et al. 2020) fine-tuned on the ANLI dataset (Nie et al., 2020); neutral and contradicting pairs were classified as non-entailing (Honovich et al., 2022b). We consider citations part of the summary's token vocabulary, but exclude them when evaluating summary quality (see Section 4 for more details on evaluation).

To check whether the citation quality transfers to other domains, we also compare our blueprint models in a zero-shot transfer setting using a recently collated benchmark called ALCE (Gao et al., 2023). The main motivation behind ALCE was to examine the citation abilities of LLMs. The benchmark contains evaluation sets from ASQA (Stelmakh et al., 2022), which focuses on ambiguous queries and long-form answers, ELI5 (Fan et al., 2019), which focuses on how/why/what questions that require explanations, and QAMPARI (Rubin et al., 2022) where the answers are lists of entities. These datasets were selected such that the answers are factual, long-form, and require information from multiple sources. For these datasets we use the passages provided by the ALCE benchmark with no further reranking. We present more details on ALCE and representative examples in Appendix D.

**Evaluation Metrics** We evaluate the quality of the summaries and additionally for models that output citations whether these are correct. Following Narayan et al. (2023), we measure the summary's *relevance* by comparing it against gold-standard responses, using ROUGE-L (Lin, 2004). The metric

computes the longest common subsequence between generated and reference text. We perform this comparison after eliminating any citations present in the response.

We also quantify the extent to which generated summaries are *faithful* to their input using textual entailment (Maynez et al., 2020; Falke et al., 2019; Narayan et al., 2022; Honovich et al., 2022a). Our entailment classifier is trained on the the Adversarial NLI dataset (Nie et al., 2020); for each sentence in the summary, we check wether it is entailed by the input passages (and report the average across all sentences to obtain an overall score). Again, we remove citations from the output to measure faithfulness.

In addition, we evaluate *citation quality* by checking whether the citations mention the *right* passages. We extend AutoAIS (Bohnet et al., 2022), a recently proposed metric for automatically measuring the attribution of short answers (in a QA setting). For long-form responses, we take sentences with in-line citations and check whether these are entailed by all the passages they cite (Amplayo et al., 2023). We then report the proportion of entailed sentences. We use the same ANLI classifier (Nie et al., 2020) described above.

For blueprint models, we would additionally expect the plan to convey information relayed by the input passages. There is no point in introducing an intermediate representation if it does not faciliate grounding to the input. We measure *blueprint quality*, by leveraging an answerability classifier (see Appendix B for details) that checks whether a question is answered by a passage. We report the proportion of questions in the blueprint that are answered by the passages it references. Questions in the blueprint cite passages *implicitly* through their respective summary sentences. Recall that by construction there is a one-to-one correspondence between blueprint questions and summary sentences (which in turn cite input passages).

## 5 AQuAMuSe Experiments

### 5.1 Comparison Models

To examine how attribution and planning interact we compare several model instantiations: a sequence-to-sequence (Section 3.1) baseline which does not have a blueprint or perform attribution (see $-$Blueprint $-$Attribution in Table 1); a model which does not have a blueprint (Section 3.1) but nonetheless performs attribution ($-$Blueprint $+$Attribution); and its mirror image, i.e., a model with a blueprint (abstractive or extractive) but no attribution ($+$Blueprint$_{\mathcal{A}|\mathcal{E}}$ $-$Attribution); and finally a blueprint model with attribution ($+$Blueprint$_{\mathcal{A}|\mathcal{E}}$ $+$Attribution). In all cases we fine-tune a LongT5 model (Guo et al. 2022, 3B parameters).

| Retrieval-based Models | | ROUGE-L | ANLI |
|---|---|---|---|
| $-$Blueprint | $-$Attribution | 63.80 | 87.20 |
| $-$Blueprint | $+$Attribution | 63.72 | 86.73 |
| $+$Blueprint$_{\mathcal{A}}$ | $-$Attribution | 64.17 | 87.88 |
| $+$Blueprint$_{\mathcal{E}}$ | $-$Attribution | 72.25 | 87.92 |
| $+$Blueprint$_{\mathcal{A}}$ | $+$Attribution | 63.49 | **88.05** |
| $+$Blueprint$_{\mathcal{E}}$ | $+$Attribution | **72.98** | 88.01 |

Table 1: AQuAMuSe results: summary quality.

| Retrieval-based Models | | Answerability | AutoAIS |
|---|---|---|---|
| $-$Blueprint | $+$Attribution | — | 72.64 |
| $+$Blueprint$_{\mathcal{A}}$ | $+$Attribution | 92.27 | 74.16 |
| $+$Blueprint$_{\mathcal{E}}$ | $+$Attribution | **97.97** | **74.35** |

Table 2: AQuAMuSe results: attribution quality.

The blueprint models presented by Narayan et al. (2023) constitute the state of the art on AQuA-MuSe. Obtaining a ROUGE score of 61.43 with a LongT5 model without planning, while one of their blueprint variant obtained ROUGE score of 59.24 and ANLI of 83.37. However, their models vary from ours in that they do not perform attribution or use a retrieval step to find relevant passages; they instead rely on the long encoding context of LongT5. The blueprints are also slightly different in that they use a list of question-answer pairs, while we focus on questions only.

### 5.2 Results and Analysis

Table 1 presents our results on AQuAMuSe when evaluating summary relevance and faithfulness. In Table 2 we evaluate attribution quality for the models which output citations. Overall, we observe that the extractive blueprint model ($+$Blueprint$_{\mathcal{E}}$ $+$Attribution) performs best across evaluation metrics.

| Abstractive Blueprint | | Answerability | AutoAIS |
|---|---|---|---|
| In-line Citations | $b_1 \ldots b_k \quad s_1 \, [c] \ldots s_m \, [c]$ | 92.27 | 74.16 |
| Adding Question Citations | $Q_1 \colon b_1 \ldots Q_k \colon b_k \quad s_1 \, [Q_1, c] \ldots s_m \, [Q_k, c]$ | 92.95 | 60.86 |
| Blueprint In-line Citations | $b_1 \, [c] \ldots b_k \, [c] \quad s_1 \ldots s_m$ | 93.67 | 61.49 |
| **Extractive Blueprint** | | **Answerability** | **AutoAIS** |
| In-line Citations | $b_1 \ldots b_k \quad s_1 \, [c] \ldots s_m \, [c]$ | 97.97 | 74.35 |
| Implicit Citations | $b_1 \ldots b_k \quad s_1 \ldots s_m$ | 98.20 | 63.79 |

Table 3: Results for Blueprint models with different citation formats. Where $b_i$ is a blueprint question, $s_i$ is a sentence in the output summary and $c$ is the set of citations $C_i$ for the corresponding sentence.

In terms of ROUGE-L, it improves the state of the art by 10 points, while it also improves faithfulness (see ANLI column). We believe this is due to the exceptionally high blueprint attribution quality of 97.97% which further suggests that an attributable plan helps improve the quality of the summary. Both blueprint models ($+\text{Blueprint}_{\mathcal{A}|\mathcal{E}}$ $+$Attribution) improve AutoAIS over a model which performs attribution without a plan ($-$Blueprint $+$Attribution), which reinforces the argument that blueprints improve attribution quality.

Our models are overall better compared to Narayan et al. (2023), achieving a new state of the art in the AQuAMuSe dataset. However, as mentioned earlier, the two approaches are not directly comparable as they differ in terms of the input they expect and blueprint definition. Nonetheless, our results show that blueprint models are more faithful (in terms of ANLI) without sacrificing summary quality (ROUGE-L). In contrast, Narayan et al. (2023) report a tradeoff between faithfulness and summarization quality. We show example output in Appendix E. We further analyze the behavior of the blueprint models by examining the degree to which they copy text from the input, the quality of their citations, and how attribution and planning interact.

**Abstractiveness** We investigate the hypothesis that blueprint models might perform better simply because it is easier for them to copy sentences or passages from the input. In Figure 2 we evaluate the proportion of unique $n$-grams in the generated summary (compared to the input passages) for different $n$-gram sizes (where $n = \{3, 5, 10, 20, 40, 80\}$); we examine three attribution models: a baseline without blueprint, and the abstractive/extractive blueprint models. The plot reveals that the extractive model generates the most abstractive responses, while the no-blueprint baseline copies longer chunks of text from the input. This means that blueprints help models consolidate information from multiple sources, while still being faithful to the input.

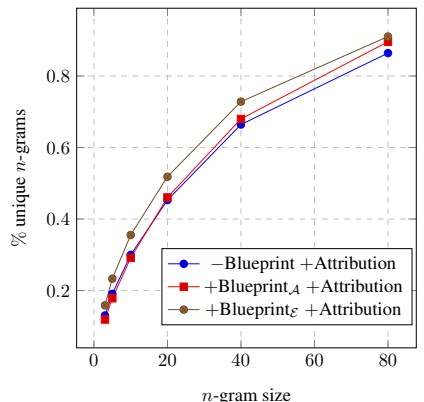

Figure 2: Proportion of unique $n$-grams.

**Citation Format** The expressivity afforded by the blueprint plans further allows us to define citations in different ways. Our experiments have so far adopted in-line citations as a common format for all models (with or without blueprints). We now examine whether alternative formats have any bearing on the performance of blueprint models by: (a) augmenting in-line passage citations with references to blueprint questions; (b) placing in-line citations in the blueprint (after each question) rather than in the output summary; and (c) having implicit citations for extractive models only; in this case, we do not generate any citations but attribution can be inferred on the basis of the questions being aligned to summary sentences and copied from the input passages. These formats introduce a more explicit connection between passage citations and blueprint questions, so we expect the blueprint attribution quality to increase. Figure 3 shows the same example summary from Figure 1 with different citation formats.

We present our results with varying citation formats in Table 3, where we indeed see a slight increase in answerability for blueprints, however, at the expense of AutoAIS.

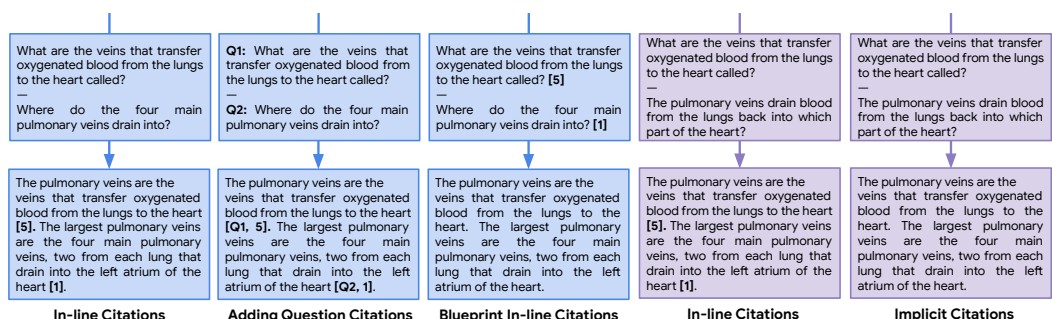

Figure 3: Examples of different citation formats for blueprint models. The top blocks correspond to the blueprint and the bottom ones to the output summary. Abstractive blueprint formats are colored blue while the extractive blueprint formats are colored purple.

**Grounding and Controllability**   Finally, we examine whether the *planning* aspect of the blueprint models remains intact when they are also expected to generate summaries with references to the input. We measure the extent to which the output is grounded to the blueprint plan using the answerability classifier. Specifically, we check if the questions in the blueprint are answerable by the summary as a whole and by indivual sentences (and report the average). Table 4 shows our results for blueprint models with and without attribution. As can be seen, extractive blueprint models are less grounded compared to their abstractive counterparts, even though the former are better at attribution (see Table 2).

We further illustrate how to improve the attribution quality of abstractive models, therby showcasing the controllability aspect of blueprint models. We simply remove (post-hoc) generated blueprint questions that are unanswerable based on the input passages, and then generate the summary using the filtered blueprint. This post-processing substantially improves attribution by 7.09 AutoAIS points (compared to Table 2) while retaining or marginally improving output quality on other dimensions (63.12 ROUGE-L, 89.45 ANLI, 93.08, Answerability, and 81.25 AutoAIS).

| Retrieval-based Models | Summary | Sentences |
|---|---|---|
| +Blueprint$_{\mathcal{A}}$ − Attribution | 96.67 | 96.49 |
| +Blueprint$_{\mathcal{E}}$ − Attribution | 91.32 | 80.06 |
| +Blueprint$_{\mathcal{A}}$ + Attribution | 97.46 | 93.64 |
| +Blueprint$_{\mathcal{E}}$ + Attribution | 91.22 | 79.97 |

Table 4: Grounding results for Blueprint Models (with and without attribution).

## 6   ALCE EXPERIMENTS

### 6.1   COMPARISON MODELS

We next examine the generalization capability of our approach on out-of-domain examples. We compare our abstractive blueprint model (+Blueprint$_{\mathcal{A}}$, +Attribution) with the LLM-based pipelines reported in Gao et al. (2023). We chose the abstractive blueprint model over the extractive variant in this experiment, under the assumption that a question generation model is not available at transfer time (and as a result we cannot obtain questions in input passages for the extractive model).

Gao et al. (2023) report experiments with two base LLMs, namely ChatGPT (`gpt-3.5-turbo-0301`) and LLaMA (Touvron et al., 2023), both of which were prepended with detailed instructions and a few demonstrations for in-context learning. Their ChatGPT pipelines include (a) Vanilla, an end-to-end model where the input is the query and top-5 passages; (b) Summarization condeses the passages first with a separate LLM and thus can process more of them; (c) Snippet extracts snippets from passages using an LLM; (d) Inline Search, where instead of providing passages in the input, the LLM calls search in-line whenever needed; and (e) Closed Book, where no passages are provided and attribution happens post-hoc (i.e., for each sentence, find the best passage to cite). We did not include systems that additionally use response reranking and interaction strategies because they are orthogonal improvements and do not consistently improve performance. LLaMA-based pipelines are versions of the Vanilla approach described above that use either (f) LLaMA-13B, or

|  |  | ASQA | | ELI5 | | QAMPARI | | Average | |
|---|---|---|---|---|---|---|---|---|---|
|  |  | Correct. | Attrib. | Correct. | Attrib. | Correct. | Attrib. | Correct. | Attrib. |
| ChatGPT | Vanilla (5-psg) | 40.4 | 73.0 | 12.0 | 50.5 | 20.8 | 20.7 | 24.4 | 48.1 |
|  | Summarization (10-psg) | **43.3** | 65.2 | 12.3 | 49.8 | 22.3 | **24.6** | 26.0 | 46.5 |
|  | Snippet (10-psg) | 41.4 | 61.1 | 14.3 | 47.5 | 22.9 | 23.9 | 26.2 | 44.2 |
|  | Inline Search | 32.4 | 58.2 | **18.6** | 44.6 | 18.7 | 14.9 | 23.2 | 39.3 |
|  | Closed Book | 38.3 | 26.7 | **18.6** | 15.5 | **24.7** | 10.0 | **27.2** | 17.4 |
|  | LLaMA-13B | 26.9 | 12.6 | 3.9 | 3.9 | 9.4 | 6.9 | 13.4 | 7.8 |
|  | Vicuna-13B | 31.9 | 50.6 | 10.0 | 17.4 | 14.9 | 12.9 | 18.9 | 27.0 |
|  | LongT5 3B (10-psg) +Blueprint$_\mathcal{A}$ +Attribution | 33.8 | **77.8** | 5.2 | **60.9** | 12.9 | 10.8 | 17.3 | **49.8** |

Table 5: Results on ALCE benchmark for LLM-based pipeline models Gao et al. (2023) and proposed abstractive blueprint model. Definitions for correctness and attribution are in Section 6.2.

(g) Vicuna-13B (Chiang et al., 2023), which is essentially LLaMA further fine-tuned on user-shared conversations.[3]

## 6.2 EVALUATION METRICS

Following Gao et al. (2023), we evaluate system output in terms of correctness and attribution. Correctness is measured differently for each dataset; for ASQA they follow the original paper (Stelmakh et al., 2022) and calculate the recall of correct short answers; for ELI5, they use InstructGPT (`text-davinci-003`; Ouyang et al., 2022) to generate sub-claims from the target response, and an NLI model (TRUE; Honovich et al., 2022b) to predict whether the response entails each sub-claim; for QAMPARI, they follow the original paper (Rubin et al., 2022) and calculate precision and recall by checking the exact match to the gold-standard list of answers, considering recall to be 100% if the prediction includes at least five correct answers; here, we only use recall to measure correctness since precision is calculated over a list of predicted answers and our models output only one long answer (given the AQuAMuSe training). Attribution is evaluated using citation recall, which determines if the output is entirely supported by cited passages, and citation precision, which identifies any irrelevant citations. For metrics that have both precision and recall numbers, for the sake of brevity, we report the F1 score.

## 6.3 RESULTS

Our results are summarized in Table 5. On average, we observe that the blueprint model is better than competing models in terms of attribution. We posit that this is due to the fact that it has been explicitly fine-tuned with outputs that have citations, which in turn improves attribution, rendering the model robust across datasets. In terms of correctness, our model on average performs better than LLaMA-13B but worse than the other models, which is not surprising as it is applied to new datasets without access to in-domain training. In contrast, the comparison LLM-based pipelines have been exposed to few-shot demonstrations.

## 7 CONCLUSION

In this paper we focused on retrieval augmented generation with citations. We explored the attribution capabilities of plan-based models and proposed different variants depending on how the plans are created (i.e., abstractively or extractively). Our experiments revealed several interesting findings. Summary quality improves for blueprint models when these learn to generate citations, both in terms of relevance and faithfulness. Conversely, this is not the case for models without an intermediate planning stage, in fact attribution slightly hurts performance (see second line in Table 1). The attribution mechanism is also important; formulating the attribution as in-line citations seems beneficial as well as a tight alignment between the input and the blueprint questions. Finally, attribution is a transferable skill for blueprint models, across datasets and information seeking tasks.

---

[3]`https://sharegpt.com/`

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

## A    QUESTION GENERATION IMPLEMENTATION

Two question generation models were used to obtain questions for input passages and blueprints: one model was trained on SQuAD v2 (Rajpurkar et al., 2018) and another one on Natural Questions (Kwiatkowski et al., 2019b). Both question generators were built on top of a T5 3B (Raffel et al., 2020) base model, and expect either of the following input formats:

1. General-Purpose QG, where the model generates a question that can be answered anywhere in the passage:

   ```
   Generate question >>> [PASSAGE]
   ```

2. Sentence-Specific QG, where the model generates a question specific to a sentence in the passage:

   ```
   Generate question >>> [PASSAGE] >> [SENTENCE]
   ```

where `[PASSAGE]` and `[SENTENCE]` are the passage and the sentence where the answer to question (to-be generated) is found.

## B    ANSWERABILITY CLASSIFIER IMPLEMENTATION

Our answerability classifier repurposes the SQuAD v2 (Rajpurkar et al., 2018) dataset that includes unanswerable questions. Specifically, a T5 11B model is fine-tuned to accept as input a question and a passage in the following format:

```
question: [question] context: [passage]
```

and to generate `Yes` as output in cases where `[question]` can be answered given `[passage]` as context and `No` otherwise.

## C    BLUEPRINT SELECTION EXAMPLES

### C.1    ABSTRACTIVE BLUEPRINTS

Table 6 shows a small-scale example of how questions in abstractive blueprints are selected. Given the three-sentence summary in the example, we generate 3 SQuAD questions (marked with the $Sq$ subscript) and 3 NQ questions (marked with the $N$ subscript) for each sentence. Additionally, we generate 3 SQuAD and 3 NQ summary-level questions, giving us a total of 24 questions. Note that in our experiments, we use 10 SQuAD/NQ questions instead of three. For each sentence, we then select the most lexically overlapping question from the union of the sentence-specific questions and the summary-level questions. It may therefore happen that the selected question is from the summary-level question set (e.g., the question for $S_1$ is selected from the summary-level set). Finally, we join these questions together with a bar in between to create the blueprint.

### C.2    EXTRACTIVE BLUEPRINTS

Table 7 shows a small-scale example of how questions in extractive blueprints are selected. For each of the five input passages (10 in our experiments), we generate the same amount of questions using the method described above. That is, for each passage we generate sentence-level and summary-level questions (similar to what is shown in Table 6) using both the SQuAD and the NQ question generation models. We then greedily select five questions for each passage, such that they have the highest lexical overlap with the passage and minimal overlap with each other. We ensure this constraint by removing the already overlapping lexicons from the passage after greedily selecting a question. Once we have the passage questions, we are ready to select the questions in the blueprint. We first filter out questions that are not answerable using the answerability classifier (AF column in the table). We then select, for each sentence in the summary, the question from the filtered list of question that has the highest lexical overlap with the sentence (SS column in the table). Finally, we join these questions together with a bar in between to create the blueprint.

| Questions from Summary Sentences | SS |
|---|---|
| Sentence 1 Questions | |
| $Q_{Sq,1}$: What is the purpose of pulmonary veins? | |
| $Q_{Sq,2}$: What are the pulmonary veins? | |
| $Q_{Sq,3}$: What veins transfer oxygenated blood from the lungs to the heart? | |
| $Q_{N,1}$: where does the pulmonary vein carry blood to | |
| $Q_{N,2}$: where does the blood in the pulmonary vein come from | |
| $Q_{N,3}$: where do the pulmonary veins carry blood to | |
| Sentence 2 Questions | |
| $Q_{Sq,1}$: How many pulmonary veins are there? | |
| $Q_{Sq,2}$: How many pulmonary veins are in each lung? | |
| $Q_{Sq,3}$: Where do the four main pulmonary veins drain into? | $S_2$ |
| $Q_{N,1}$: where does the blood in the pulmonary vein go | |
| $Q_{N,2}$: where does the pulmonary vein carry blood to | |
| $Q_{N,3}$: where do the pulmonary veins connect to the heart | |
| Sentence 3 Questions | |
| $Q_{Sq,1}$: What are pulmonary veins a part of? | |
| $Q_{Sq,2}$: The pulmonary veins are a part of what? | |
| $Q_{Sq,3}$: What circulation are the pulmonary veins a part of? | $S_3$ |
| $Q_{N,1}$: where does the pulmonary vein carry blood to | |
| $Q_{N,2}$: where do the pulmonary veins carry blood to | |
| $Q_{N,3}$: where does the blood in the pulmonary vein go | |
| Summary-Level Questions | |
| $Q_{Sq,1}$: What are the veins that transfer oxygenated blood from the lungs to the heart called? | $S_1$ |
| $Q_{Sq,2}$: What are the veins that transfer oxygenated blood from the lungs to the heart? | |
| $Q_{Sq,3}$: How many main pulmonary veins are there? | |
| $Q_{N,1}$: where does the pulmonary vein carry blood to | |
| $Q_{N,2}$: where does the blood in the pulmonary vein go | |
| $Q_{N,3}$: where do the pulmonary veins carry blood to | |
| Summary | |
| [The pulmonary veins are the veins that transfer oxygenated blood from the lungs to the heart.]$_{S_1}$ [The largest pulmonary veins are the four main pulmonary veins, two from each lung that drain into the left atrium of the heart.]$_{S_2}$ [The pulmonary veins are part of the pulmonary circulation.]$_{S_3}$ | |

Table 6: Abstractive Blueprint selection for an example summary. Each sentence has its own list of sentence-specific questions, and there is another list of general-purpose questions that all sentences can select from. Each sentence can only select one question to create We split the summary into sentences and select no more than one question per sentence (see SS column).

## D  THE ALCE BENCHMARK

ALCE (Gao et al., 2023) is a collection of datasets aimed at evaluating LLM citation capabilities. They contain factual questions which require long-form answers aggregating information over multiple sources (i.e., 100-word passages). We summarize various statistics on these datasets in Table 8 and describe them below.

**ASQA** (Stelmakh et al., 2022) focuses on ambiguous questions which have multiple interpretations (see Table 9 for an example). The questions were taken from AmbigQA (Min et al., 2020), while long-form answers were crowdsourced by synthesizing information from multiple documents. For ALCE (Gao et al., 2023), ASQA questions were additionally paired with Wikipedia passages (2018-12-20 snapshot) which purportedly contained the answers.

**QAMPARI** (Rubin et al., 2022) is an open-domain QA dataset where answers are lists of entities, drawn from different passages. All questions in QAMPARI have at least 5 answers, with an average of 13 answers. Multi-answer questions were automatically generated using manually defined templates, answers were collated from Wikipedia, and examples were verified and paraphrased by crowdworkers (see Table 9 for an example). For ALCE (Gao et al., 2023), QAMPARI questions were also paired with Wikipedia passages (2018-12-20 snapshot).

**ELI5** (Fan et al., 2019) mostly constists of how/why/what questions that require in-depth long answers and multiple passages as evidence. Questions (and answers) were elicited from the subreddit *Explain Like I'm Five* (ELI5) where users are encouraged to provide answers which are comprehensible by a five year old (see Table 9). For ALCE (Gao et al., 2023), ELI5 questions were paired with passages from Sphere (Piktus et al., 2021), a filtered version of Common Crawl.

ALCE (Gao et al., 2023) contains 1,000 randomly selected examples from each dataset (development set). It does not provide training data as it is aimed at assessing the citation capabilities of LLMs.

| Questions from Input Passages | AF | SS |
|---|---|---|
| **Passage 1 Questions** | | |
| $Q_1$: what is essential for bringing oxygenated blood from the lungs back to the heart | ✓ | |
| $Q_2$: the pulmonary veins drain blood from the lungs back into which part of the heart | ✓ | $S_2$ |
| $Q_3$: the pulmonary arteries connect the heart to what | | |
| $Q_4$: what connects the heart to the lungs and allows the blood to get oxygenated | | |
| $Q_5$: what drains blood from the lungs back into the left atrium of the heart | ✓ | |
| **Passage 2 Questions** | | |
| $Q_1$: in the heart-lung circulation, what brings deoxygenated blood from the right side of the heart to the lungs to be oxygenated | | |
| $Q_2$: where does the pulmonary artery bring blood back to | | |
| $Q_3$: how is oxygenated blood drained back to the heart | ✓ | |
| **Passage 3 Questions** | | |
| $Q_1$: what happens to oxygenated blood when it is pumped through arteries to other parts of the body | | |
| $Q_2$: what holds the secret to appropriate functioning of the circulatory and cardiovascular system | ✓ | |
| $Q_3$: what is the function of pulmonary veins in the heart | ✓ | |
| $Q_4$: the body can not live and grow without what | ✓ | |
| $Q_5$: pulmonary veins carry blood from which part of the body | ✓ | |
| **Passage 4 Questions** | | |
| $Q_1$: which pulmonary veins pass behind the right atrium and superior vena cava; the left in front of the descending thoracic aorta | | |
| $Q_2$: what are large blood vessels that carry oxygenated blood from the lungs to the left atrium of the heart | ✓ | |
| $Q_3$: the number of pulmonary veins opening into the left atrium can vary between three and how many in the healthy population | ✓ | |
| $Q_4$: the superior pulmonary vein lies in front of and a little below which artery | | |
| $Q_5$: what is the anterior surface of the bronchus invested by | | |
| **Passage 5 Questions** | | |
| $Q_1$: what are the veins that transfer oxygenated blood from the lungs to the heart called | ✓ | $S_1$ |
| $Q_2$: what is a main function of the pulmonary veins | ✓ | |
| $Q_3$: how many primary pulmonary veins does each lung have | ✓ | |
| $Q_4$: where do the pulmonary veins drain into the heart | ✓ | |
| $Q_5$: what circulation are the pulmonary veins a part of | ✓ | $S_3$ |
| **Summary** | | |
| [The pulmonary veins are the veins that transfer oxygenated blood from the lungs to the heart.]$_{S_1}$ [The largest pulmonary veins are the four main pulmonary veins, two from each lung that drain into the left atrium of the heart.]$_{S_2}$ [The pulmonary veins are part of the pulmonary circulation.]$_{S_3}$ | | |

Table 7: Extractive Blueprint selection for an example summary. Each passage has at most five questions. We first use the answerability classifier to filter out questions that are not answerable by the summary (see AF column). We then select at most one question for each sentence in the summary based on their lexical overlap (see SS column).

| Dataset | Question Type | #passages | Question Length | Answer Length |
|---|---|---|---|---|
| ASQA | Factoid (ambiguous) | Wikipedia (21M) | 12.5 | 65.7 |
| QAMPARI | Factoid (list) | Wikipedia (21M) | 10.5 | 13.0 |
| ELI5 | Why/How/What | Sphere (899M) | 17.4 | 120.6 |

Table 8: Datasets in the ALCE benchmark (Gao et al., 2023). Question and Answer Length refer to the average number of words; for QAMPARI, Answer Length is the average number of answers. Examples are shown in Table 9.

# E   EXAMPLE OUTPUT

Tables 10, 12, and 14 present output examples for the baseline system (−Blueprint +Attribution) and our two blueprint variants (+Blueprint$_{\mathcal{A}|\mathcal{E}}$ +Attribution). Blueprints created by the plan-based systems are respectively shown in Tables 11, 13, and 15. In general, we observe that the baseline system considers a smaller number of input passages when generating its response, while plan-based summaries contain more diverse passage refrences. As far as the blueprints are concerned, we find that the abstractive system generates (on average) more questions compared to the extractive model and the reference. Although the questions making up the blueprints of the two models are not identical, we see that they cover similar topics. For instance, in Table 11 the blueprints emphasize topics like the mean temperatures of a tropical monsoon climate, its precipitation, dry seasons, and variance in temperature.

| ASQA |
| --- |
| Q: Who played Bill Weasley in Harry Potter and the Prisoner of Azkaban? |
| A: The Weasely family is a family of wizards including Molly and Arthur Weasely as well as their seven children. Richard Fish appeared as Bill briefly in the film adaptation of Harry Potter and the Prisoner of Azkaban. Domhnall Gleeson, the son of actor Brendan Gleeson (Alastor Moody in the series), played Bill Weasley in Harry Potter and the Deathly Hallows. Chris Rankin played Percy Weasely and Fred and George Weasely were played by identical twins James Phelps and Oliver Phelps. Rupert Grint played Ron Weasely. |
| QAMPARI |
| Q: Who won a Coke Zero Sugar 400 competition? |
| A: Sam McQuagg, Bobby Allison, Fireball roberts, Jimmie Johnson, Tony Stewart, Tim Richmond, Jeff Gordon, David Ragan, Erik Jones, Tony Stewart, Jamie McMurray, Kyle Busch, Cale Yarborough, Dale Earnhardt Jr., Brad Keselowski, David Pearson, Dale Jarrett, Jeff Gordon, Tony Stewart, Greg Biffle, A. J. Foyt, Richard Petty, Justin Haley, Tony Stewart. |
| ELI5 |
| Q: How does so much of our trash end up in the ocean? |
| A: Because water flows downhill and very often ends up in rivers which very often end up in oceans. So when it rains, trash is washed downhill and into streams and rivers and ultimately the ocean. |

Table 9: Examples of question-answer pairs from ALCE datasets; input passages are omitted for the sake of brevity.

| describe the seasonal patterns of the tropical savanna biome |
|---|
| [0] However, unlike most tropical climates, a sizeable amount of precipitation falls during the dry season. In essence, this version of the monsoon climate generally has less pronounced dry seasons than tropical savanna climates . . . |
| [1] Tropical monsoon climates have monthly mean temperatures above 18 °C in every month of the year and feature wet and dry seasons, as Tropical savanna climates do. Tropical monsoon climates however features its driest month seeing on average less than 60 mm and this latter fact is in direct contrast to a tropical savanna climate . . . |
| [2] 64. Tropical wet and dry or savanna climate (Aw) have a pronounced dry season, with the driest month having precipitation less than 60 mm and less than 1/25 of the total annual precipitation. Example(s) of areas with this climate is(are) . . . |
| [3] Tropical monsoon climate, occasionally also known as a tropical wet climate or tropical monsoon and trade-wind littoral climate in climate classification, is a relatively rare type of climate. Tropical monsoon climates have monthly mean temperatures above 18 °C (64 °FF) . . . |
| [4] The second letter indicates the precipitation pattern—w indicates dry winters (driest winter month average precipitation less than one-tenth wettest summer month average precipitation. s indicates at least three times as much rain in the wettest month of winter as in the driest month of summer . . . |

| −Blueprint +Attribution |
|---|
| Tropical monsoon climate, occasionally also known as a tropical wet climate or tropical monsoon and trade-wind littoral climate in climate classification, is a relatively rare type of climate. [3] Tropical monsoon climates have monthly mean temperatures above 18 °C (64 °F) in every month of the year and feature wet and dry seasons, as Tropical savanna climates do. [3] Unlike tropical savanna climates however, a tropical monsoon climate's driest month sees less than 60 mm (2.36 in) of precipitation. [3] Also a tropical monsoon climate tends to see less variance in temperatures during the course of the year than a tropical savanna climate. [3] This climate has a driest month which nearly always occurs at or soon after the winter solstice for that side of the equator. [3] |

| +Blueprint$_\mathcal{A}$ +Attribution |
|---|
| Tropical monsoon climate, occasionally also known as a tropical wet climate or tropical monsoon and trade-wind littoral climate in climate classification, is a relatively rare type of climate. [3] Tropical monsoon climates have monthly mean temperatures above 18 °C (64 °F) in every month of the year and feature wet and dry seasons, as tropical savanna climates do. [3] Unlike tropical savanna climates however, a tropical monsoon climate's driest month sees less than 60 mm (2.36 in) of precipitation. [3] Additionally, a tropical monsoon climate tends to see less variance in temperatures during the course of the year than a tropical savanna climate. [1] This climate has a driest month which nearly always occurs at or soon after the "winter" solstice for that side of the equator. [3] There are generally two versions of a tropical monsoon climate: Less pronounced dry seasons and Extraordinarily rainy wet seasons and pronounced dry seasons. [0] Regions with this variation of the monsoon climate typically see copious amounts of rain during the wet season. [1] |

| +Blueprint$_\mathcal{E}$ +Attribution |
|---|
| Tropical monsoon climate, occasionally also known as a tropical wet climate or tropical monsoon and trade-wind littoral climate in climate classification, is a relatively rare type of climate. [3] Tropical monsoon climates have monthly mean temperatures above 18 °C (64 °F) in every month of the year and feature wet and dry seasons, as tropical savanna climates do. [1] Unlike tropical savanna climates however, a tropical monsoon climate's driest month sees less than 60 mm (2.36 in) of precipitation. [3] Also a tropical monsoon climate tends to see less variance in temperatures during the course of the year than a tropical savanna climate. [3] This climate has a driest month which nearly always occurs at or soon after the "winter" solstice for that side of the equator. [3] |

| Reference Summary |
|---|
| Tropical savanna climate or tropical wet and dry climate is a type of climate that corresponds to the Köppen climate classification categories "Aw" and "As". [9] Tropical savanna climates have monthly mean temperatures above 18 °C (64 °F) in every month of the year and typically a pronounced dry season, with the driest month having precipitation less than 60 mm and also less than 100 − (total annual precipitation (mm) / 25) of precipitation. [9] This latter fact is in direct contrast to a tropical monsoon climate, whose driest month sees less than 60 mm of precipitation but has more than 100 − (total annual precipitation (mm) / 25) of precipitation. In essence, a tropical savanna climate tends to either see less rainfall than a tropical monsoon climate or have more pronounced dry seasons. [0] |

Table 10: Example responses to an AQuAMuSe query (top) for a sequence-to-sequence model which does not include plans, two blueprint models, (abstractive and extractive), and gold standard summary. The second block in the table shows the 5 best retrieved passages in abridged form (only the first sentence is given). Passage questions are also omitted for the sake of brevity.

| Abstractive Blueprint |
| --- |
| Q: what is a tropical monsoon climate occasionally also known as? |
| Q: what are the monthly mean temperatures of a tropical monsoon climate? |
| Q: what is the average precipitation in a tropical monsoon climate's driest month? |
| Q: a tropical monsoon climate tends to see less variance in temperatures during the course of the year than what other type of climate? |
| Q: what month nearly always occurs at or soon after the winter solstice for that side of the equator? |
| Q: how many versions of a tropical monsoon climate are there? |
| Q: what do regions with this variation of the monsoon climate typically see during the wet season? |
| **Extractive Blueprint** |
| Q: what type of climate has monthly mean temperatures above 18 °C (64 °F) in every month of the year? |
| Q: what type of climate sees less than 60 mm of precipitation in its driest month? |
| Q: a monsoon climate tends to see more of what than a tropical savanna climate? |
| Q: what type of climate has less pronounced dry seasons than tropical monsoon climates? |
| Q: what do regions with less pronounced dry seasons typically see during the wet season? |
| **Reference Blueprint** |
| Q: what is the driest month in a tropical wet and dry or savanna climate? |
| Q: what type of climate has monthly mean temperatures above 18 °C (64 °F) in every month of the year? |
| Q: a tropical monsoon climate's driest month sees less than what amount of precipitation? |
| Q: what do tropical wet and dry or savanna climates have a pronounced dry season? |

Table 11: Abstractive and extractive blueprints for model summaries shown in Table 10. For comparison, we also include the blueprint of the reference summary.

| difference between regular file and pipe file in unix |
| --- |
| [0] Tytower, I think I know what's the reason of your problem. In old Linux (a.k.a. Old Kerbel version) named pipes weren't created by mkfifo command . . . |
| [1] It is not possible to change the backup destination for third-party tools. For this destination, the only objects created in the file system are named pipes. Named pipes occupy no space in the file system . . . |
| [2] The recovery process ends successfully, so I suppose that system correctly read all necessary recovery parameters from backup catalog. Still, it is confusing looking at the wrong log backup location in generated SQL recovery statement. . . . |
| [3] SAP HANA supports certified third party backup tools via the backint interface. Backups are transferred via pipe from the SAP HANA database to the 3rd party backup agent, which runs on the SAP HANA database server and then sends the backups to the 3rd party backup server. . . . |
| [4] This project shows you how to implement your own on-line radio receiver in Linux, using as major basis the mplayer application . . . |
| $-$Blueprint $+$Attribution |
| In Unix-like operating systems, a file is a regular file, while a pipe is a named pipe. [0] |
| $+$Blueprint$_\mathcal{A}$ $+$Attribution |
| Named pipes are a special type of file system object in Unix-like computer systems. [0] All Unix-like computer systems include at some point named pipes. [0] |
| $+$Blueprint$_\mathcal{E}$ $+$Attribution |
| In Unix-like operating systems, a file is either a regular file or a named pipe. [0] Named pipes are special files which occupy no space in the file system and can be created and destroyed like any other file. [1] |
| Reference Summary |
| In Unix-like operating systems, a file is a regular file, while a pipe is a named pipe. [0] |

Table 12: Example responses to an AQuAMuSe query (top) for a sequence-to-sequence model which does not include plans, two blueprint models, (abstractive and extractive), and gold standard summary. The second block in the table shows the 5 best retrieved passages in abridged form (only the first couple of sentences are given). Passage questions are also omitted for the sake of brevity.

| Abstractive Blueprint |
| --- |
| Q: what are named pipes a special type of in unix-like computer systems? |
| Q: what do all unix-like computer systems include at some point? |
| Extractive Blueprint |
| Q: what type of files are created when a backup is made to a pipe? |
| Q: named pipes occupy how much space in the file system? |
| Reference Blueprint |
| Q: named pipes are special files that can exist where on the file system? |
| Q: named pipe special files are made with the command mkfifo as in what ? |

Table 13: Abstractive and extractive blueprints for model summaries shown in Table 12. For comparison, we also include the blueprint of the reference summary.

| briefly explain html. what are the essential features of html |
| --- |
| [0] HTML is Hyper Text Markup Language – the standard markup language for creating web applications and pages, and a cornerstone of internet programming. HTML denotes the building blocks of a web page, such as text, paragraphs, hyperlinks, images, and other items. . . |
| [1] HTML: Hyper Text Markup Language is the standard markup language used to create webpages and web applications. HTML represents the structure of a web page with semantic information, images, interactive forms and other objects. . . |
| [2] HyperText Markup Language or simply HTML is the markup language used by web developers to create web pages. It is written in the format of HTML elements containing tags bracketed in angle brackets like <html> . . . |
| [3] HTML: HyperText Markup Language, commonly referred to as HTML, is the standard markup language used to create web pages. HTML is a standardized system for tagging text files to achieve font, color, graphic, and hyperlink effects on World Wide Web pages. . . |
| [4] HTML - HyperText Markup Language, commonly referred to as HTML, is the standard markup language used to create web pages. Web browsers can read HTML files and render them into visible or audible web pages. HTML describes the structure of a website semantically along with cues for. . . |
| $+\text{Blueprint}_\mathcal{A}$ +Attribution |
| HyperText Markup Language, commonly referred to as HTML, is the standard markup language used to create web pages. [4] Web browsers can read HTML files and render them into visible or audible web pages. [3] Web browsers do not display HTML tags and scripts, but use them to interpret the content of the page. [2] HTML describes the structure of a website semantically along with cues for presentation, making it a markup language, rather than a programming language. [4] HTML elements form the structure blocks of all web pages. [2] It allows images and objects to be embedded and can be used to create interactive forms. [2] |
| $+\text{Blueprint}_\mathcal{E}$ +Attribution |
| HTML is HyperText Markup Language – a standard markup computer language for representing the contents of a page of hypertext. [0] Web browsers receive HTML documents from a server and render them into visible or audible web pages. [4] HTML describes the structure of a web page semantically and originally included cues for the appearance of the document. [8] |
| Reference Summary |
| Hypertext Markup Language (HTML) is the standard markup language for creating web pages and web applications. [8] With Cascading Style Sheets (CSS) and JavaScript it forms a triad of cornerstone technologies for the World Wide Web. [5] Web browsers receive HTML documents from a web server or from local storage and render them into multimedia web pages. [2] HTML describes the structure of a web page semantically and originally included cues for the appearance of the document. [8] |

Table 14: Example responses to an AQuAMuSe query (top) for a sequence-to-sequence model which does not include plans, two blueprint models, (abstractive and extractive), and gold standard summary. The second block in the table shows the 5 best retrieved passages in abridged form (only the first couple of sentences are given). Passage questions are also omitted for the sake of brevity.

| Abstractive Blueprint |
| --- |
| Q: what is the standard markup language used to create web pages? |
| Q: what can web browsers read and render into visible or audible web pages? |
| Q: what do web browsers use to interpret the content of a page? |
| Q: what describes the structure of a website semantically along with cues for presentation? |
| Q: what do html elements form the structure blocks of? |
| Q: what can be embedded in html and used to create interactive forms? |
| Extractive Blueprint |
| Q: what is the standard markup language for creating web applications and pages? |
| Q: what is a standardized system for tagging text files to achieve font, color, graphic, and hyperlink effects on world wide web pages? |
| Q: web browsers receive html documents from a server and render them into what? |
| Q: html describes the structure of a website semantically along with cues for what? |
| Reference Blueprint |
| Q: what is the standard markup language for creating web pages and web applications? |
| Q: along with html and css, what is a cornerstone technology for the world wide web? |
| Q: web browsers receive html documents from where? |
| Q: html originally included cues for the appearance of what? |

Table 15: Abstractive and extractive blueprints for model summaries shown in Table 14. For comparison, we also include the blueprint of the reference summary.

