# OpenReview forum: "Learning to Plan and Generate Text with Citations"
_ICLR.cc/2024/Conference — Submitted to ICLR 2024_

### Official Review · Reviewer_eZwk · 2023-10-30

**Soundness:** 3 good
**Presentation:** 3 good
**Contribution:** 3 good
**Rating:** 6
**Confidence:** 4

**Summary:**

The paper proposes a set of questions related to the input as an intermediate step for long-form QA and summarization tasks. The authors experiment with different ways to generate the intermediate set of questions and show that the model performs better than similar works in the literature in summary quality and attribution/citations in the summary. They further test the generalizability of the model on a new task and comparing it with other LLM pipelines.

**Strengths:**

Within the proposed work the authors have performed comparisons on different variations.
On attribution, the proposed work performs better than larger models on new domains.

**Weaknesses:**

1. The description in the problem formulation is incomplete. I had some struggles understanding the difference btw abstractive and extractive models. Figure 1 is good, the authors would use the figure to clearly explain the input, output, and how is the blueprint being used/generated.
2. There is no apple-apples comparison in Table 1.
- The authors could test their model without Retriever to make it comparable with Narayan et al easily.
- I think the proposed model can be extended to a setting with QA pairs similar to Narayan et al.
These two experiments will help us understand the exact contribution of the difference in blueprint style and the modifications proposed compared to the one in the literature (Narayan et al.)
3. A simple/traditional baseline for attribution quality will better ground our understanding.
4. Table 5 with (-blueprint + attributions) will make for better comparison

**Questions:**

No questions

---

> ### Author Response · Authors · 2023-11-21
> **We thank the reviewer for their useful feedback! We address the points of inclarity below:**
>
> > The description in the problem formulation is incomplete. I had some struggles understanding the difference btw abstractive and extractive models. Figure 1 is good, the authors would use the figure to clearly explain the input, output, and how is the blueprint being used/generated.
>
> We are glad that Figure 1 helped with the understanding, we have added additional references to it when presenting the model.
>
> > There is no apple-apples comparison in Table 1.
> The authors could test their model without Retriever to make it comparable with Narayan et al easily.
> I think the proposed model can be extended to a setting with QA pairs similar to Narayan et al. These two experiments will help us understand the exact contribution of the difference in blueprint style and the modifications proposed compared to the one in the literature (Narayan et al.)
>
> Indeed, the bottom part of Table 1 corresponded to the results from Narayan et al. which as we stated (Section 5.1 second paragraph) are not directly comparable. We only included them here for completeness. It is certainly not the goal of this paper to have a better blueprint model compared to the ones proposed by Narayan et al., rather we want to explore how to extend these models to perform attribution.
>
> During the early stages of this research we did explore the difference between using only questions plans or QA plans, we did not find meaningful differences (as shown below for an abstractive end to end model), so we decided to continue using Q plans since these are more flexible (i.e., generated questions can be generic and do not require an answer span to be extracted from the input).
>
> |    | ROUGE-L | ANLI  |
> |----|---------|-------|
> | QA | 0.558   | 0.760 |
> | Q  | 0.553   | 0.769 |
>
> We also studied the impact of using the retriever (numbers below for an abstractive model). We found that the reranker does indeed improve both ROUGE and ANLI. However, the main motivation for using the reranker is because we need a set of passages on which to perform attribution (which is not the case in Narayan et. al where they use the full documents in the input)
>
> |    | ROUGE-L | ANLI  |
> |----|---------|-------|
> | Q     |     0.553      | 0.769 |
> | Q + reranker |     0.623      | 0.791 |
>
> > A simple/traditional baseline for attribution quality will better ground our understanding.
>
> The baseline model we trained (Model (a) in Figure 1) is the traditional attribution baseline.
>
> > Table 5 with (-blueprint + attributions) will make for better comparison
>
> In Section 6,  our goal is to examine whether the most performant model from the main experiments (Section 5) learns robust attribution capabilities that generalize to domain and task changes. That is why we only evaluated the Blueprint model.

---

> > ### Author Response · Authors · 2023-11-23
> > **Friendly reminder**
> >
> > Given these clarifications would you consider increasing your score?

---

### Official Review · Reviewer_8rJ6 · 2023-11-01

**Soundness:** 3 good
**Presentation:** 2 fair
**Contribution:** 2 fair
**Rating:** 3
**Confidence:** 4

**Summary:**

This paper explores the attribution capabilities (generating citations) of the plan based methods for long-form Question Answering where plans are framed as an intermediate step of generating sequence of questions aka blueprints. Two kinds of models (extractive and abstractive) are proposed to generate these blueprints and empirical results show improved performance on both AQuAMuSe and ALCE datasets using LongT5 as the base model.

**Strengths:**

- The paper introduces an interesting approach to incorporating attribution (i.e generating citations) into text generation models through blueprints, which can potentially improve the trustworthiness and quality of generated content.
- This work also evaluates model performance in transfer settings, demonstrating the robustness and generalization capabilities of the blueprint models.

**Weaknesses:**

- The paper is hard to follow, referencing other papers and models without providing detailed contextual information, making it challenging for readers unfamiliar with these works to fully understand the content.
- It is also not clear what are the main contributions of the model wrt Narayan et al (2023). It seems that the main differences are just the formulation of the blueprints as a set of questions rather than the Question-Answer pairs and re-ranking of passages before generation. It would help to clearly specify this in the beginning of the text.
- It is not clear if the passages used for grounding are retrieved or just re-ranked from the given 10 passages for each instance.
- Table 1 suddenly shows ablations with attribution while it is not explicitly explained in the earlier sections. Does the attribution here refer to “generating text with citations” only?  Was this originally not part of the task formulation? If not, then do the authors post-process the generated summary to remove the citations while performing evaluation? It would help to clarify this in the starting of the text.
- It is not clear as to how extractive models learn to copy. Is there any utilization of the forced copy mechanism? The authors mention that a question generation model is not available at transfer time to ALCE dataset. It is not clear why this is not the case for an abstractive model?
- Pictorial depiction of the models could aid in understanding the proposed approach.
- It seems the relevant baselines haven’t been used like GPT-3.5/4 or open-sourced LLMs like LLaMA-2 where the field has evolved a lot in recent years.
- Human evaluation is not provided. Reliability of automatic eval metrics for NLG is a known issue in the community.
- Table 5 shows consistent decreased performance of the proposed models for correctness compared to the baselines (especially on QAMPARI and ELI5 datasets). It indicates that the model learns to hallucinate, producing more citations but incorrect results.


Even though the results are promising at the initial level, the paper can be improved in terms of writing clarity and structure. It would be interesting to see a revised version of the paper in future conferences.

**Questions:**

- From Table 1, it seems that vanilla seq2seq (line 1) performs better than all the models of Narayan et al. which uses blueprints. This seems to oppose the earlier observations of Narayan et al. Do the authors have an intuition behind this? Also how would this model differ from the base LongT5 model of Narayan et al?
- Are the number of contextual passages 10 for all the query examples? It would be nice to see an ablation as to how the models perform with increasing complexity of the context.
- Could the authors comment on the individual performance of Question Generator and Answerability classifier? What is the overall accuracy of the classifier? How many generated questions were actually determined to be answerable by the classifier?
- In section 5.2, the authors note that “The plot reveals that the extractive model generates the most abstractive responses”. Do authors have an intuition of this counterintuitive observation?
- It is mentioned that blueprint plans improve attribution quality. Could the authors quantify this? It is hard to distinguish between attribution and blueprint.
- Fig 1 shows the same summaries for the baseline as well as extractive, abstractive models which seem to be scripted and less informative. Actual outputs of different models would be interesting.


Suggestions/Comments:
Page 5, Section 3.2:  sentencss -> sentences

---

> ### Author Response · Authors · 2023-11-21
> **We thank the reviewer for their feedback! We address their questions below**
>
> > It is also not clear what are the main contributions of the model wrt Narayan et al (2023).
>
> In this paper we explore how to best develop models with a *built-in* mechanism for attribution, which is not something that was addressed in Narayan et al. 2023. We investigate whether blueprint-style models can enhance attribution capabilities compared to alternatives which do not follow a plan-based architecture. A secondary question relates to the blueprints themselves and how they should be formulated for better attribution (i.e., abstractive vs extractive blueprints). In total, this work addresses three questions (articulated in the last paragraph of the introduction) in relation to attribution, which again is out of scope for Narayan et al. 2023. Compared to related work on attribution, we are not aware of any other models which employ planning to enhance evidence-based generation capabilities  (See updated second to last paragraph in the Introduction)
>
> > It is not clear if the passages used for grounding are retrieved or just re-ranked from the given 10 passages for each instance.
>
> Just re-ranked (Section 4. Second paragraph)
> > Table 1 suddenly shows ablations with attribution while it is not explicitly explained in the earlier sections. Does the attribution here refer to “generating text with citations” only? Was this originally not part of the task formulation? If not, then do the authors post-process the generated summary to remove the citations while performing evaluation? It would help to clarify this in the starting of the text.
>
> We have rewritten  our description of  comparison models in Section 5.1 and added  references to previous sections.
>
> The attribution was explained earlier in the paper (Section 3.1), so yes “attribution” in Table 1 refers to generating text with citations. Indeed attribution is part of the task formulation, nevertheless by also showing results for models without attribution we can observe whether this added capability improves answer quality.
>
> To evaluate answer quality (ROUGE-L and ANLI) , we indeed remove  citations (we now state this explicitly in Section 4).
> > It is not clear as to how extractive models learn to copy. Is there any utilization of the forced copy mechanism?
>
> The model *learns* to copy given that it is fine-tuned on  data which contains blueprint questions  directly copied from the input. We verified this fact by checking the percentage of copied questions in the predictions which is 98.94% exact match or 99.91% ROUGE-L.
> > The authors mention that a question generation model is not available at transfer time to ALCE dataset. It is not clear why this is not the case for an abstractive model?
>
> The abstractive model does not need the input to be augmented with extra questions (as opposed to the extractive model) and it has already been trained to generate a plan (i.e., a sequence of questions) and the answer, so we can use it directly with any other input and it will generate the blueprint and the answer. If we wanted to use the extractive model, we would need to generate questions for the input passages, as this model expects that as part of its input (See Figure 1).
> > Pictorial depiction of the models could aid in understanding the proposed approach.
>
> Do you mean to add something else apart from Figure 1 ?
> > It seems the relevant baselines haven’t been used like GPT-3.5/4 or open-sourced LLMs like LLaMA-2 where the field has evolved a lot in recent years.
>
> We do compare  our approach against these models using the ALCE benchmark.
> > Table 5 shows consistent decreased performance of the proposed models for correctness compared to the baselines (especially on QAMPARI and ELI5 datasets). It indicates that the model learns to hallucinate, producing more citations but incorrect results.
>
> Firstly, we found a typo in our ALCE results and updated the QAMPARI results of our model (from 0.8 to 12.9). With this fix, we now see that across all datasets the correctness score of our blueprint model is on par with LLaMA-13B. Even still, the reviewers are correct in pointing out that the correctness of our model is inferior compared with the others LLMs.
>
> We emphasize that this is *not due to hallucinations*, but rather because the models returned *faithful but incorrect* responses. Given that the attribution score is high, information in the responses should be attributed back to the passages (see examples in Tables 10, 12, and 14). This is in contrast to LLaMA-13B, which has a similar correctness score but a significantly worse attribution score.
>
> Furthermore, the main goal of evaluating on ALCE is comparing the robustness of the attribution capabilities of our blueprint model (3B) to that of large language models (13B) and commercial solutions (ChatGPT). Our results show that such capabilities are robust and superior to all the other models.

---

> > ### Author Response · Authors · 2023-11-21
> > **We address the remaining questions below**
> >
> > > Human evaluation is not provided. Reliability of automatic eval metrics for NLG is a known issue in the community.
> >
> > Our work aims to improve attribution capabilities of generative models. As a result, we focus primarily on attribution-related evaluation metrics.  We agree that automatic metrics such as AutoAIS are not completely reliable; we thus performed a human elicitation study where 5 English speakers labeled attributed sentences as having a correct or incorrect citation (i.e. the sentence is entailed by the passage it cites).  We evaluated  the output of the baseline model and the two blueprint variants (extractive and abstractive) on 10 randomly selected examples.  The attribution accuracy obtained per model was:
> > - Baseline: 0.72
> > - Abstractive: 0.854
> > - Extractive: 0.923
> >
> > Which is in line with the automatic evaluations presented in the paper.
> >
> > > From Table 1, it seems that vanilla seq2seq (line 1) performs better than all the models of Narayan et al. which uses blueprints. This seems to oppose the earlier observations of Narayan et al. Do the authors have an intuition behind this? Also how would this model differ from the base LongT5 model of Narayan et al?
> >
> > The models from Narayan et al. were included to give an idea of the state of the art (when attribution is not taken into account) but are not  comparable  (see Section 5.1 second paragraph); Narayan et al. did not use a reranker for the input passages, and their blueprints consist of QA pairs rather than questions only. The goal of this paper is not to have better models than Narayan et al., but rather to instill attribution capabilities in blueprint models and explore whether these are superior to vanilla attribution models without planning.
> >
> > > Are the number of contextual passages 10 for all the query examples? It would be nice to see an ablation as to how the models perform with increasing complexity of the context.
> >
> > Yes, all the examples have 10 passages in the input. Indeed it could be interesting, but we believe it is not directly related to our main questions around attribution with plan based models.
> >
> > > Could the authors comment on the individual performance of Question Generator and Answerability classifier? What is the overall accuracy of the classifier?
> >
> > The Answerability classifier has a 92.5% accuracy on the development set of the SQuAD v2 dataset. Regarding the Question Generator, we were not able to measure performance during training but as shown in the examples in the appendix (Table 7) and when evaluating the answerability of questions we saw that the Question Generator produced good quality questions.
> >
> > >How many generated questions were actually determined to be answerable by the classifier?
> >
> > In Table 2 we see the percentage of generated questions that are answerable, 97% in the case of the extractive model and 92% with the abstractive model.
> > > In section 5.2, the authors note that “The plot reveals that the extractive model generates the most abstractive responses”. Do authors have an intuition of this counterintuitive observation?
> >
> > We discuss this in Section 5.2, last line in the “Abstractiveness'' paragraph. Blueprints help models consolidate information from multiple sources (in Figure 2 both blueprint models are more abstractive than the baseline). Furthermore a better blueprint would help the abstractiveness of the output even more, and we know that the extractive model generates better blueprints as measured in Table 2 by the Answerability score.
> >
> > > It is mentioned that blueprint plans improve attribution quality. Could the authors quantify this? It is hard to distinguish between attribution and blueprint.
> >
> > In Table 2 we make this direct comparison (quantifying the attribution quality with AutoAIS) we see that rows 2 and 3 (blueprint models) have higher AutoAIS scores than row 1 (the model without a blueprint plan).
> >
> > > Fig 1 shows the same summaries for the baseline as well as extractive, abstractive models which seem to be scripted and less informative. Actual outputs of different models would be interesting.
> >
> > We show actual outputs of the models in Appendix E

---

> > > ### Comment · Reviewer_8rJ6 · 2023-11-22
> > > **Acknowledgement of the response**
> > >
> > > Thank you authors for the response!
> > >
> > > - It seems that the preliminary human study does not support the results well with only 10 examples. There is also limited information about the human eval which seems to be rushed. How was the human evaluation conducted with some information about their qualification? Was the inner agreement calculated among annotators? The procedure of annotation should be described in detail. It is also not clear what dataset was used for human evaluation.
> > > - The authors mention that “Table 1 results are not comparable to Narayan et al”. Is it because of the re-ranking? An ablation study in this regard without the retriever could provide direct comparison with the relevant baselines.
> > > - Similarly, while recent LLMs have been compared for the ALCE benchmark, how well do they perform on AQuAMuSe dataset as a baseline.
> > > - As another fellow reviewer pointed out, the difference between abstractive and extractive models is still not clear. The authors mention that the extractive model “learns” to copy. Do the extractive models provide spans or generate the questions without any copy supervision? How does this make it different from abstractive models?
> > > - The technical novelty is limited with contributions beyond attribution are not extensively detailed where planning for generation is not new (Narayan et al). It seems that this work is incremental in this regard.
> > >
> > > With these questions in mind, I invite the authors to include relevant baselines uniformly for both AQuAMuSe and ALCE benchmark with detailed ablation studies of the importance of the attribution and blueprint for both benchmarks. It would also be nice to have a proper human evaluation study. I stand by my original rating till then.

---

> ### Author Response · Authors · 2023-11-23
>
> Thank you again for responding and acknowledging our response!
>
> * **Human Study**
>
> We apologize for the lack of clarity of the human study we conducted during this rebuttal period. The annotators were given a citing sentence in the summary and the passage/s it has cited (they also have access to the whole summary as context in case the sentence needed to be decontextualized). They were then asked to classify whether all information in the citing sentence can be found in the set of cited passage/s. The score we presented is the accuracy of their classification.
>
> Given the setting above, the study has been done at the **sentence level**, therefore while we may have only asked annotators to do 10 examples, this actually meant them annotating 153 sentences in total. Still, we do agree with the reviewer that this may not be enough. However, we hope the reviewer understands that this is all due to the time constraint. To make the study more reliable, we additionally tested significance (McNemar’s test; p<0.05) and it shows that the extractive and baseline models are significantly different.
>
> We plan to include a more extensive human evaluation in the final version.
>
> * **Ablation with retriever**
>
> We want to remind the reviewer that the main goal of the paper is to improve attribution through planning. The main reason why Narayan et al.'s models are not comparable is because these models **do not produce attribution**. However, we do understand that the reviewer is curious of the differences between our models and theirs. We responded the following to another reviewer who also asked a similar question:
>
> During the early stages of this research we did explore the difference between using only questions plans or QA plans, we did not find meaningful differences (as shown below for an abstractive end to end model), so we decided to continue using Q plans since these are more flexible (i.e., generated questions can be generic and do not require an answer span to be extracted from the input).
>
> |          |   ROUGE-L  | ANLI  |
> | -- | -- | -- |
> | QA   |    0.558      | 0.760 |
> | Q     |     0.553      | 0.769 |
>
> We also studied the impact of using the retriever (numbers below for an abstractive model). We found that the reranker does indeed improve both ROUGE and ANLI. However, the main motivation for using the reranker is because we need a set of passages on which to perform attribution (which is not the case if we simply pass all the documents in the input as done in Narayan et. al).
>
> |                            | ROUGE-L | ANLI  |
> | -- | -- | -- |
> | Q                      |     0.553      | 0.769 |
> | Q + reranker |     0.623      | 0.791 |
>
> * **LLMs on AQuAMuSE**
>
> We did an additional experiment for another reviewer where we used a 5-shot prompt on GPT-3.5 turbo with 16k context (gpt-3.5-turbo-16k) and obtained the following results on 100 examples from the test split of AQuAMuSE (again, apologies if we could not do the larger test split).
>
> |                          		          | AutoAIS |
> |---------------------------|----------------------|
> | -Blueprint +Attribution       | 28.11               	           |
> | +Blueprint_A +Attribution | 52.21            	           |
>
> The results still show that the blueprint contributes to having better attribution but the numbers are overall lower than those of fine tuned models.
>
> We have also tested on the larger test split using an in-house LLM and got similar results, however due to time constraints, we are not able to obtain legal approvals.
>
> * **Extractive Blueprint Model**
>
> We believe the reviewer's questions regarding the extractive blueprint model can be answered by Figure 1. There is no built-in extractive mechanism (i.e., no copy mechanisms or special training/loss/supervision). The difference from abstractive model is the addition of passage-specific questions in the input (please see Figure 1 of the paper). The extractive model learns to copy from these questions in the input, while the abstractive model does not have access to questions in the input. We do not need any copy mechanism as mentioned in our previous response since the questions in the output are 98.94% of the time copied from the input (and has 99.91% ROUGE-L compared to the input).

---

### Official Review · Reviewer_LX3E · 2023-11-01

**Soundness:** 4 excellent
**Presentation:** 4 excellent
**Contribution:** 4 excellent
**Rating:** 8
**Confidence:** 4

**Summary:**

This paper develops and motivates a method for long-form question answering with citations using a variant of the blueprint method of Narayan et al. 2023. In the present paper, blueprint models take as input a query and a set of retrieved passages for that query, and then the model predicts a blueprint (sequence of quetions) and a long-for response to the questions (with citations) in a single decoding step. The paper reports on trained blueprint models that do extractive answering and abstractive answering. The main experiments are with the AQuAMuSe dataset, and both extractive and abstractive blueprint models perform extremely well. The models also perform exceptionally well at the citation step, and a separate set of experiments on the ALCE dataset show that the AQuAMuSe-trained models are competitive with ChatGPT and other LLMs.

**Strengths:**

This is an interesting paper that addresses a really important and challenging problem -- answering complex questions with citations. The citation piece is an excellent step towards enabling people to verify for themselves what LLMs seem to be telling them.

The paper itself is easy to read, and the experimental results are rich and interesting.

**Weaknesses:**

I don't have weaknesses to point out per se. I am supportive of publishing the paper.

**Questions:**

The ALCE experiments are interesting and informative. It looks like a trained blueprint model is effective here. My question is about whether the blueprint strategy might be effective with pure in-context learning, with a model like GPT-3.5. This would be similar to SelfAsk, I suppose. My apologies of this was already done; the description at the bottom of page 8 is very high-level and so I can't really tell what the prompting strategies were here.

---

> ### Author Response · Authors · 2023-11-21
> **We thank the reviewer for their feedback!  Below we address the question stated:**
>
> >The ALCE experiments are interesting and informative. It looks like a trained blueprint model is effective here. My question is about whether the blueprint strategy might be effective with pure in-context learning, with a model like GPT-3.5.
>
> We have indeed experimented with pure in-context learning, but unfortunately it does not work as well as a fine tuned model. For instance, a 5-shot prompt on GPT-3.5 turbo with 16k context (gpt-3.5-turbo-16k) obtained the following results on 100 examples from the test split of Aquamuse.
> |                          		          | AutoAIS |
> |---------------------------|----------------------|
> | --Blueprint +Attribution       | 28.11               	           |
> | +Blueprint_A +Attribution | 52.21            	           |
> We still see that the blueprint contributes to having better attribution but the numbers are overall lower than those of fine tuned models.

---

> > ### Comment · Reviewer_LX3E · 2023-11-21
> > **Thanks!**
> >
> > Thank you – a valuable piece of evidence!

---

### Official Review · Reviewer_vp7e · 2023-11-05

**Soundness:** 3 good
**Presentation:** 2 fair
**Contribution:** 3 good
**Rating:** 6
**Confidence:** 4

**Summary:**

This paper studies query-based summarization with citations. The authors propose to create blueprints - generating a set of questions to guide the generation of summaries with citations. The authors propose two methods for blueprint generation: the abstractive model and the extractive model.

Experimental results demonstrate the effectiveness of the proposed framework on both the summary quality and the citation quality - the blueprint generation and the attribution generation mutually enhance each other as well as the summary generation quality.

The authors also conduct experiments on the ALCE benchmark. The proposed method achieved better performance for citation generation while falling behind for correctness. Since the ALCE benchmark does not have training data, the proposed method cannot surpass the LLM-based in-context learning methods in terms of correctness, but surpass citation generation since it's explicitly fine-tuned.

**Strengths:**

The proposed method is decent and achieves good performance, and the proposed abstraction and extractive modules are demonstrated performant.

**Weaknesses:**

1. The major concern is the lack of novelty in the proposed method. The question generation modules are not new.
2. In the experiments on the ALCE benchmark, the correctness score falls far behind the LLM-based few-shot prompting methods.
3. The writing of the paper should be significantly improved. There are too many redundant narratives but less focus on the overall technical contribution.

**Questions:**

Are you going to release the code for better reproducibility?

---

> ### Author Response · Authors · 2023-11-21
> **We thank the reviewer for their feedback! Below we address the points raised as weaknesses**
>
> > 1. The major concern is the lack of novelty in the proposed method. The question generation modules are not new.
>
> In this paper we explore how to best develop models with a *built-in* mechanism for attribution, which is not something that was addressed in Narayan et al. 2023. We investigate whether blueprint-style models can enhance attribution capabilities compared to alternatives which do not follow a plan-based architecture. A secondary question relates to the blueprints themselves and how they should be formulated for better attribution (i.e., abstractive vs extractive blueprints). In total, this work addresses three questions (articulated in the last paragraph of the introduction) in relation to attribution, which again is out of scope for Narayan et al. 2023. Compared to related work on attribution, we are not aware of any other models which employ planning to enhance evidence-based generation capabilities  (See updated second to last paragraph in the Introduction)
>
> > 2. In the experiments on the ALCE benchmark, the correctness score falls far behind the LLM-based few-shot prompting methods.
>
> Firstly, we found a typo in our ALCE results and updated the QAMPARI results of our model (from 0.8 to 12.9). With this fix, we now see that across all datasets the correctness score of our blueprint model is on par with LLaMA-13B. Even still, the reviewers are correct in pointing out that the correctness of our model is inferior compared with the others LLMs.
>
> We emphasize that this is *not due to hallucinations*, but rather because the models returned *faithful but incorrect* responses. Given that the attribution score is high, information in the responses should be attributed back to the passages (see examples in Tables 10, 12, and 14). This is in contrast to LLaMA-13B, which has a similar correctness score but a significantly worse attribution score.
>
> Furthermore, the main goal of evaluating on ALCE is comparing the robustness of the attribution capabilities of our blueprint model (3B) to that of large language models (13B) and commercial solutions (ChatGPT). Our results show that such capabilities are robust and superior to all the other models.
>
> > 3. The writing of the paper should be significantly improved. There are too many redundant narratives but less focus on the overall technical contribution.
>
> We have uploaded an updated version which makes the contributions clearer, and includes several rewrites following the reviewers’ comments and questions.

---

> > ### Author Response · Authors · 2023-11-23
> > **Friendly reminder**
> >
> > Given these clarifications would you consider increasing your score?

---

### Meta-Review · Area_Chair_AhLc · 2023-12-06

**Metareview:**

This paper explores query-based summarization with citations.  Blueprints (i.e. a set of questions) are created and used to guide propose to the generation of summaries with citations, given a query and several relevant passages. Two methods are proposed for producing blueprints: the abstractive model and the extractive model. Experimental results demonstrate the effectiveness of the proposed framework on both the summary quality and the citation quality.

Strengths:  The problem investigated in the paper is interesting, and the idea of using blueprints for summary generation with citations sounds reasonable. The proposed method achieves good performance on automatic metrics.

Weaknesses: The novelty of the presented method is limited. Planning with LLMs is not new, and similarly generating text with citations is not a novelty. The proposed method borrows many techniques/implementations from Narayan et al (2023), and the contributions made in this paper need to be presented more clearly. Presentation of the current submission can still be improved to properly differentiate from the prior works. More relevant baselines need to be considered. A comprehensive human evaluation is necessary to support the work.

Considering the weaknesses outweigh the strengths, I recommend to reject this paper.

More comments:  The current (ideal) setting is that relevant passages are retrieved and provided for a query, however, in practice, the retrieved passages usually contain irrelevant/noisy ones. I wonder the performance of the proposed method in this real setting.

**Justification For Why Not Higher Score:**

Please see the weaknesses listed in the meta-review.

**Justification For Why Not Lower Score:**

N/A

---

### Decision · Program_Chairs · 2024-01-16

Reject